

# Inter-ethnic genetic variations and novel variant identification in the partial sequences of *CYP2B6* gene in Pakistani population

Sagheer Ahmed[1], Hizbullah Khan[1], Asifullah Khan[2], Muhammad Hanif Bangash[3], Abrar Hussain [4], Mughal Qayum[5] and Mohammad Hamid Hamdard[6]

[1] Department of Basic Medical Sciences, Shifa College of Pharmaceutical Sciences, Shifa Tameer-e-Millat University, Islamabad, Pakistan
[2] Department of Biochemistry, Abdul Wali Khan University, Mardan, Pakistan
[3] Pakistan Institute of Nuclear Science & Technology, Islamabad, Pakistan
[4] Baluchistan University of Information Technology and Management Sciences, Quetta, Pakistan
[5] Department of Pharmacy, Kohat University of Science & Technology, Kohat, Pakistan
[6] Faculty of Biology, Kabul University, Kabul, Afghanistan

Corresponding author
Mohammad Hamid Hamdard,
hamid.hamdard@ku.edu.af

## ABSTRACT

**Background:** Some single nucleotide polymorphisms (SNPs) in the cytochrome P450 *(CYP)2B6* gene lead to decreased enzyme activity and have an impact on drug metabolism. The present study was designed to investigate the patterns of genetic distinction across a hypervariable region of the *CYP2B6* gene, known to contain important SNPs, i.e. rs4803419 and rs3745274, among five major ethnic groups of the Pakistani population.

**Methods:** Arlequin v3.5.DnaSPv6.12. and network 5 resources were used to analyze population genetic variance in the partial *CYP2B6* gene sequences obtained from 104 human samples belonging to Punjabi, Pathan, Sindhi, Seraiki and Baloch ethnicities of Pakistan. The partial CYP2B6 gene region analyzed in the current study is previously known to possess important SNPs.

**Results:** The data analyses revealed that genetic variance among samples mainly came from differentiation within the ethnic groups. However, significant genetic variation was also found among the various ethnic groups. The high pairwise Fst genetic distinction was observed between Seraiki and Sindhi ethnic groups (Fst = 0.13392, *P*-value = 0.026) as well as between Seraiki and Balochi groups (Fst = 0.04303, *P*-value = −0.0030). However, the degree of genetic distinction was low between Pathan and Punjabi ethnic groups. Some SNPs, including rs3745274 and rs4803419, which are previously shown in strong association with increased plasma Efavirenz level, were found in high frequency. Besides, a novel SNP, which was not found in dbSNP and Ensemble databases, was identified in the Balochi ethnicity. This novel SNP is predicted to affect the *CYP2B6* splicing pattern.

**Conclusion:** These results may have significant implications in Pakistani ethnicities in the context of drugs metabolized by CYP2B6, especially in Seraiki and Balochi ethnicity. The novel heterogeneous SNP, found in the present study, might lead to altered drug-metabolizing potential of *CYP2B6* and, therefore, may be implicated in non-responder phenomenon.

## INTRODUCTION

The cytochrome P450 2 (CYP2) is a major family of the cytochrome enzymes, comprising of several subfamilies that aggregated together in the form of clusters in the genome. The CYP2 subfamilies include the *CYP2A, CYP2B, CYP2F, CYP2G, CYP2S* and *CYP2T* genes (*Simonsson et al., 2003*). Among these, the *CYP2B* located at chromosome 19 of the human genome, holding nine exons encodes for 491 amino acids containing protein (*Yamano et al., 1989*). The human *CYP2B6* gene has two known loci: the functional *CYP2B6* and its non-functional pseudogene *CYP2B7*, located in the center of the *CYP2A18P* locus inside a 112 kb block (*Yamano et al., 1989*). The *CYP2B6* enzyme is involved in the metabolic activation and inactivation of several drugs including anticancer such as cyclophosphamide and ifosfamide (*Chang et al., 1993*; *Granvil et al., 1999*; *Roy et al., 1999*), antimalarial such as artemesinin (*Simonsson et al., 2003*), antiviral such as efavirenz, and antidepressant such as bupropion (*Faucette et al., 2000*). Although the total fraction of the *CYP2B6* enzyme is small as compared to the total hepatic P450 family, however, it metabolizes a vast majority of pharmaceutical drugs. In addition to 7–8% of the marketed pharmaceutical drugs, *CYP2B6* metabolizes certain exogenous and endogenous substances such as nicotine and testosterone (*Rosenbrock et al., 1999*), in conjunction with other cytochrome enzymes.

The single nucleotide polymorphisms (SNPs) in the *CYP2B6* gene may affect the expression and enzyme activity of the translated protein, resulting in significant differences in the pharmacokinetics of *CYP2B6*-metabolized drugs among individuals and ethnicities, in turn, leading to variations in efficacy and toxicity (*Desta et al., 2007*; *Nyakutira et al., 2008*; *Aurpibul et al., 2012*). Several important variants of *CYP2B6* gene, i.e. *CYP2B6*$^*$2 (C64T), $^*$3 (C777A), $^*$4 (A785G), $^*$5 (C1459T), $^*$6 (G516T and A785G), $^*$7 (G516T, A785G and C1459T) (*Lang et al., 2001*), $^*$8 (A415G) and $^*$9 (G516T) have been discovered (*Lamba et al., 2003*), besides the wildtype *CYP2B6*$^*$1 allele. About 38 protein variants of the highly polymorphic *CYP2B6* gene have been identified so far (http://www. cypalleles.ki.se/ CYP2B6.htm), (*Zanger & Klein, 2013*). Genetic variants of the enzyme result in substrate, and expression dependent functional changes.

Numerous other studies have reported that genetic variations that alter the *CYP2B6* enzyme expression result in altered drug responses (*Hesse et al., 2000*; *Coller et al., 2002*; *Lerman et al., 2002*). Studies have also shown that there are major differences in the amount of enzyme and its activity amongst different individuals. Ethnicity and sex-based differences in *CYP2B6* gene expression and splice variants have been reported previously (*Lamba et al., 2003*). For example, *CYP2B6* $^*$4 variant has been shown to result in enhanced expression and altered activity of the enzymes (*Gadel, Friedel & Kharasch, 2015*). A SNP rs3745274 (*c.516G>T)* leading to *CYP2B6*$^*$6 allele was alone responsible for aberrant splicing, resulting in a low-*CYP2B6* expression phenotype. In recent years, studies

have identified *CYP2B6\*6* in association with enhanced plasma concentrations of certain drugs (*Aurpibul et al., 2012*), including efavirenz.

Pakistan is ethnically diverse; however, little is known about the distribution of *CYP2B6* genetic variations in a country of over 210 million population. Various parts of the country possess a unique lifestyle, diverse genetic background, dietary habits, culture, and geographical environment. The five ethnicities focused on in the current study are the largest genetically and linguistically distinct ethnicities of Pakistan. Among these, the Punjabi is the largest one residing in the Punjab province. Geographically the area of Punjab is comprised of eastern Pakistan and northern India. The Sindhi ethnicity primarily resides in Sindh province located in Southeast of Pakistan. Pathan ethnic population is about 15–18% of total Pakistani population. Pathan are predominantly residing in Khyber Pakhtunkhwa province situated in northwestern region of Pakistan. Balochi people are residing predominantly in Balochistan province situated in the South and Western region of Pakistan. The Saraiki people live in Southern Punjab and Southern Khyber Pakhtunkhwa. The Saraiki people are distinguished in culture and norms from the closely residing Punjabi and Pathan ethnic groups. Previously published studies about these Pakistani population ethnic groups inferred considerable genetic variation (*Qamar et al., 2002*; *Quintana-Murci et al., 2004*; *Bergström et al., 2020*). Furthermore, distinct genetic ancestry of these groups has also been reported in worldwide population genetics project (*Li et al., 2008*). The *CYP2B6* gene contains many SNPs in addition to some copy number variations. However, only a few might alter enzyme activity or be associated with certain diseases. Keep in view this scenario, we investigated a part of *CYP2B6* from the individuals belonging from five major ethnic groups of Pakistan to infer their genetic composition across this important drug-metabolizing enzyme.

## METHODS & MATERIALS

### Sample collection and DNA extraction

The study includes 104 healthy human participants belonging to five different ethnicities of Pakistan. There were 25 samples from Pathan ethnicity, 24 from Punjabi, 20 from Sindhi, 18 from Seraiki and 17 samples were from the Baloch ethnicity. The participants self-declared their ethnicities. Ethical approval for the study was obtained from the Institutional Review Board & Ethics Committee of Shifa International Hospital/Shifa Tameer-e-Millat University, Islamabad. Consent forms, both in English and Urdu (national language) were also approved by the same committee. Urdu is understood throughout Pakistan. For DNA extraction, five ml of venous blood was collected in EDTA tubes and stored in a refrigerator. DNA extraction kits (ThermoScientific, Waltham, MA, USA) wer used to isolate the genomic DNA from the blood samples. The quantity and quality of the isolated DNA was analyzed by running it on a 1% agarose gel and by using ultraviolet spectrophotometer at 260 and 280 nm wavelength. The ratio for most samples was found in the range of 1.80 ± 0.05. Isolated genomic DNA was stored at −20 °C until further processing.

## DNA sequences and PCR amplification

The *CYP2B6* target region holding important and significant genetic variants was amplified from the purified genomic DNA. The PCR products were purified and submitted for commercial DNA sequencing services to Tisngka Biotechnology, China. Each of the samples was sequenced in triplicate (3×) for sequence validation.

## Data quality check and filtration

The quality of the DNA sequences was checked using the Staden package and Finch TV v1.4 (Geospiza, Inc; Seattle, WA, USA). High-quality sequence reads were assembled using the Lasergene v7.1 package (DNASTAR Inc UA) as described previously (*Hizbullah et al., 2020*)

## Population genetic analyses

The high-quality draft sequences were compared with the Human reference genome (GRch38) coordinates deposited in the ensemble database. The SNPs positions were identified and scanned against the genome-wide association study (GWAS) database. The GWAS database holds specific markers reported in association with a particular trait. Besides, comparative analyses were performed against different worldwide populations genetic variants data acquired from the 1000 Genomes database (*Auton et al., 2015*) using Ferret v1.1 tool. The DnaSP v6.0 package was used to calculate the sequence composition of parsimony informative sites and haplotypes. The Arlequin v3.5 software was used to find out the population genetic statistics like Fst analysis (pairwise fixation index), analysis of molecular variance (AMOVA), frequencies of haplotype, and Nei's distance '$D_A$' (to estimate the nucleotide differences among the population). The haplotypes network plot was developed using the median-joining method introduced in the NETWORK 10.0 package.

## Novel SNP assessment

The functional prediction of the novel SNP was assessed by different in silico tools, including MutationTaster-2, PolyPhen2, SIFT and PredictSNP2. These tools can predict the non-synonymous and synonymous mutation of the coding and non-coding regions. PredictSNP2 was used to find out deleterious, neutral, and unknown mutations. This resource provides a predictive score based on various parameters like combine annotation dependent depletion (CADD), deleterious annotation neural networks (DANN), computational framework for annotation and prioritization in coding and non-coding region (FunSeq2), genome-wide annotation of variants (GWAVA), fitness consequence (FitCons), and functional analysis through hidden Markov models (FATHAMM).

# RESULTS

## *CYP2B6* polymorphisms

High-quality sequenced data of *CYP2B6* was generated for 104 individuals belonging from major ethnic groups of the Pakistani population. The target region was spanning genomic coordinates, i.e. 19:41006882–41007421 of the Human reference genome GRch38.

**Table 1** Allele frequencies and annotations of the CYP2B6 polymorphism.

| S/No | References | Position | REF/ALT Alleles | Consequences | Effect in HGVS-nomenclature | Minor allele frequencies (%) | | | | |
|------|-----------|----------|-----------------|--------------|------------------------------|---------|---------|--------|---------|--------|
| | | | | | | Balochi | Punjabi | Pathan | Seraiki | Sindhi |
| 1 | rs4803419 | 19:41006887 | C/T | Intron variant | g.20589C>T | 47.058 | 45.833 | 60.000 | 55.555 | 65.000 |
| 2 | rs3745274 | 19:41006936 | G/T | Missense variant | Q172H | 47.058 | 54.166 | 48.000 | 77.777 | 35.000 |
| 3 | Novel | 19:41007072 | G/G, A | Intron variant | g.20774G>A | 5.882 | No | No | No | No |

**Notes:**
*References:* SNP Id in dbSNP database; *Position:* coordinate position in refence genome (GRch38.p12 chr:19), *REF*: reference allele; *ALT*: alternate or variant allele; *HGVS-Nomenclature:* of the Human Genome Variation Society to show the description of the sequence variant (substitution, deletion, insertion, duplication etc.)

The sequenced data completely covered the exon 4 and intron 4 and partially covered the third intron region of the *CYP2B6* gene. The data analysis in the context of human reference genome GRch38 identified total three polymorphic loci. The annotations and allele frequencies of all the three polymorphic loci are given in Table 1. The two observed SNPs, i.e. rs4803419 and rs3745274 are previously reported in association with the drug response to efavirenz and/ or abacavir treatment. A single heterozygous polymorphic locus was identified at the intronic-exonic border at position chr19: 41007072 G→A (Fig. S1). This SNP was not found in the Ensemble and dbSNP databases and may be a novel SNP. During the functional annotation analyses the SNP showed no change in the primary amino acid sequence of the protein. However, the MutationTaster-2 resource identified that this polymorphism might bring a change at the splicing site or enzyme activity, while the PredictSNP2 predicted it as a neutral variant. This heterozygous polymorphic locus was found in a Balochi individual with minimal allele frequency of 0.05882 (Table 1).

## Population structure analysis

The Fixation index (Fst) statistic was employed to investigate the genetic composition across *CYP2B6* gene among ethnic groups' samples. The high pairwise Fst genetic distinction was observed between Seraiki and Sindhi ethnic groups (Fst = 0.13392, *P*-value = 0.026) as well as between Seraiki and Balochi groups (Fst = 0.04303, *P*-value = −0.0030). However, the degree of genetic distinction was low between Pathan and Punjabi ethnic groups (Fig. 1). Moreover, when these ethnic groups were compared with the 1000 Genomes samples, the Seraiki population showed high genetic distinction (*P*-value =< 0.01) in comparison with the East Asian: Han Chinese (CHB), Japanese (JPT); South Asian: Bengali (BEB), Sri Lankan Tamil (STU), Indian Telugu (ITU), Gujrati Indian (GIH); and European populations: CEU. (Utah Residents CEPH with Northern and Western European Ancestry) (Fig. 2).

AMOVA was used to estimates the genetic deviation at a molecular level among and within various ethnic groups samples. The significance of AMOVA was assessed by 99,000 permutations (Table 2). The AMOVA model revealed that genetic diversity due to variations among-ethnic groups is 1.08%, and within-ethnic groups is 98.92% (*P*-value = 0.28615). Similarly, a variance factor (Est. Var) of 0.50731 was observed within population; however, among-ethnic groups variance factor was found low 0.00554. The among-ethnic group based genetic variance in this model was although low but still
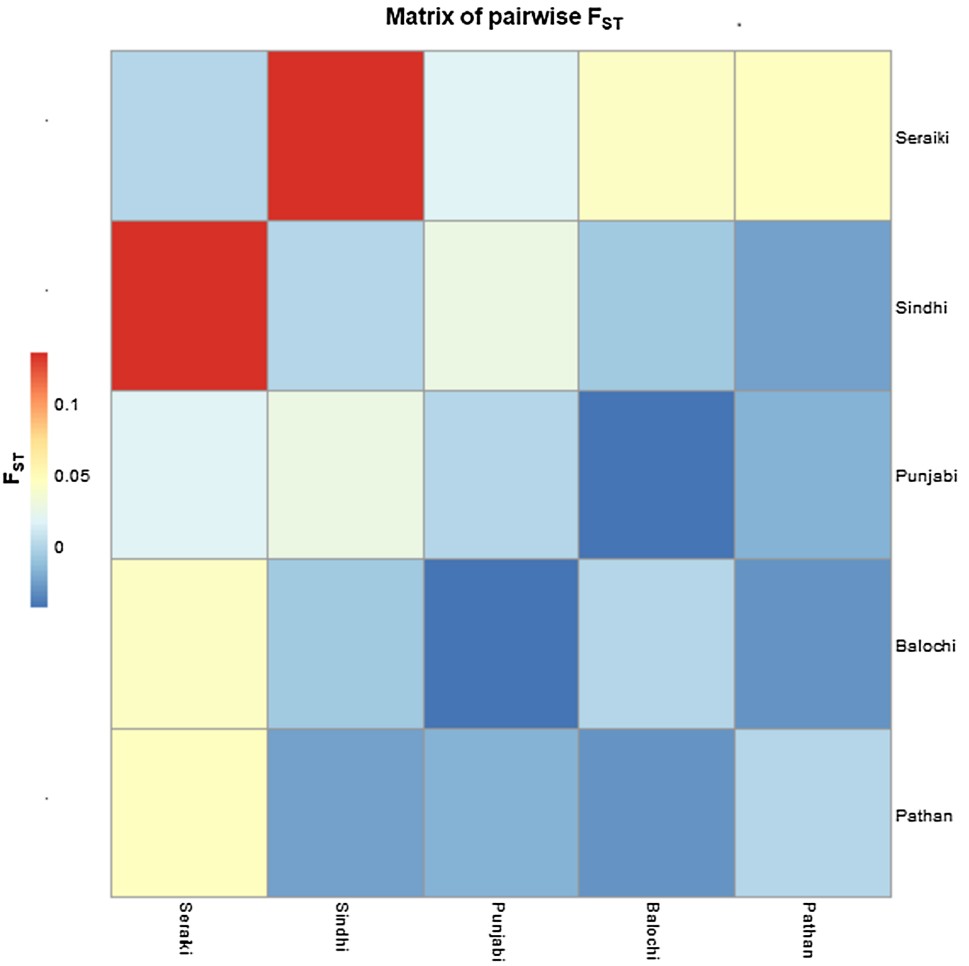

**Figure 1 Matrix of pairwise Fst-Pakistani Ethnicities.** Heatmap: Pairwise Fst (Fixation index) among five ethnic groups of Pakistani population depicted high genetic differentiation between Seraiki and Sindhi.

significant, suggesting that ethnic groups are genetically distinct. High pairwise nucleotide differences ($D_A$) were observed between Seraiki and Sindhi, Seraiki and Balochi ethnic groups.

Similarly, a high mean pairwise difference ($\pi xy$) was found in the Seraiki and Sindhi, Seraiki, and Balochi ethnic groups. The highest within-population genetic distinction (i.e., $\pi$) was observed for Balochi ethnic group samples (Fig. S2). Moreover, significant differentiation between Seraiki and other Asian/European populations was found as evident by the Nei's distance ($D_A$) during comparison of Pakistani ethnicities *CYP2B6* SNPs with additional Asian and European populations data acquired from the 1000 Genomes Project (Fig. 3).

## Haplotypes composition

A total of five haplotypes with haplotype diversity (Hd) of 0.7276 were identified for all the 104 sample sequences. The haplotype-1 was found with predominant frequency in Sindhi, Pathan, Punjabi, and Balochi samples. The haplotype-2 was also found in the same
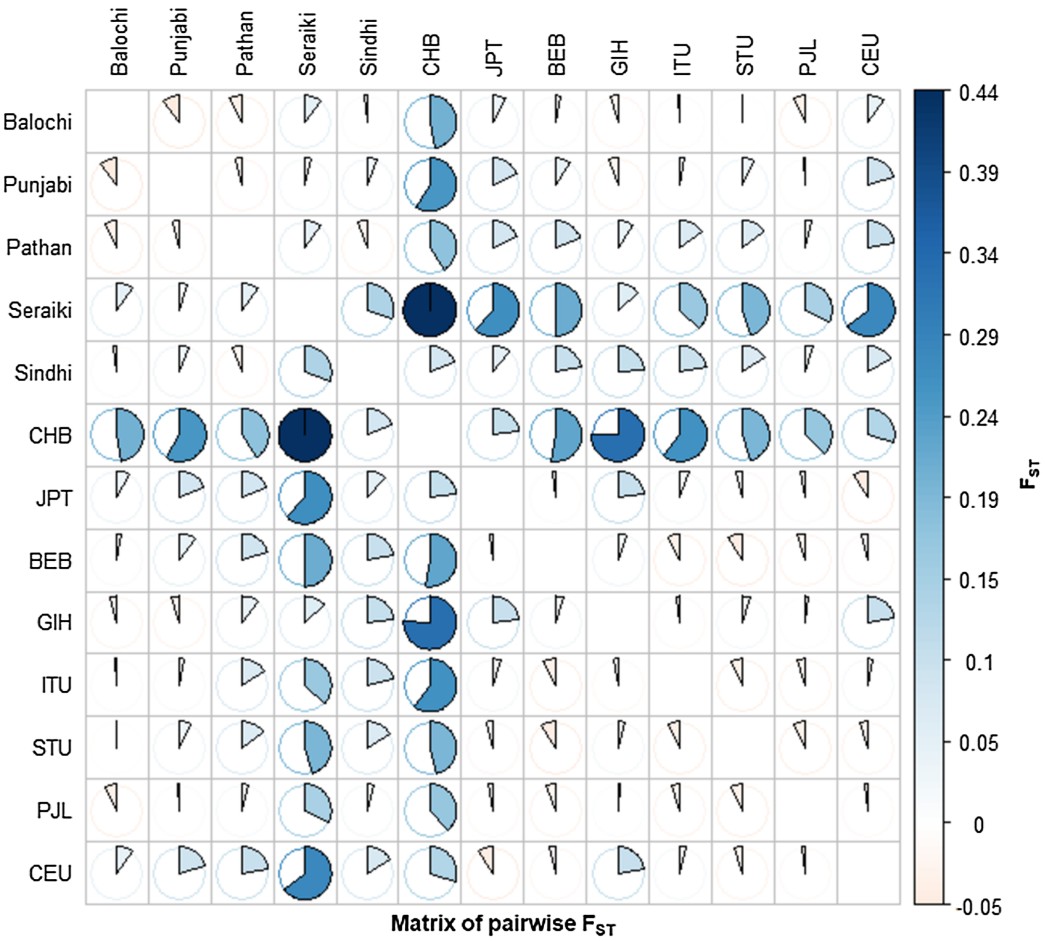

**Figure 2 Matrix of pairwise Fst-Global populations.** Pairwise Fst analysis in the Structure of other Asian Population i.e. BEB, GIH, ITU, STU, CHB, JPT and European (CEU) population data compiled from 1000 Genomes Project depicted high genetic differentiation with Seraiki individual followed by other ethnic groups.

**Table 2 Analysis of genetic distinction across CYP2B6.**

| Source of variation | Df | SS. | Est. Var | % | P-value |
|---|---|---|---|---|---|
| Among populations | 4 | 2.488 | 0.00554 | 1.08 | <0.28615 |
| Within population | 99 | 50.244 | 0.50731 | 98.92 | <0.28615 |

Notes:
*df:* degrees of freedom; *SS:* the sum of squares deviation; *Est.Var:* estimates of variance components; *%:* percentage of total variance contributed by each component.

ethnic groups including Seraiki, whereas, haplotype-3 was found at high frequency only in the Seraiki group (Fig. S3). The AMOVA analysis of the pairwise distance between haplotypes revealed that Seraiki and Sindhi predominant haplotype-2 showing substantial genetic differentiation against haplotype-5 (i.e., predominant in Balochi samples) and haplotype-1 (i.e., predominant in Sindhi and Balochi). Moreover, the haplotype-3 (i.e. predominant in Seraiki samples) showing significant genetic distinction against haplotype-5 and haplotype-4 (predominant in Balochi and Sindhi samples (Fig. 4)).
**Average number of pairwise differences**

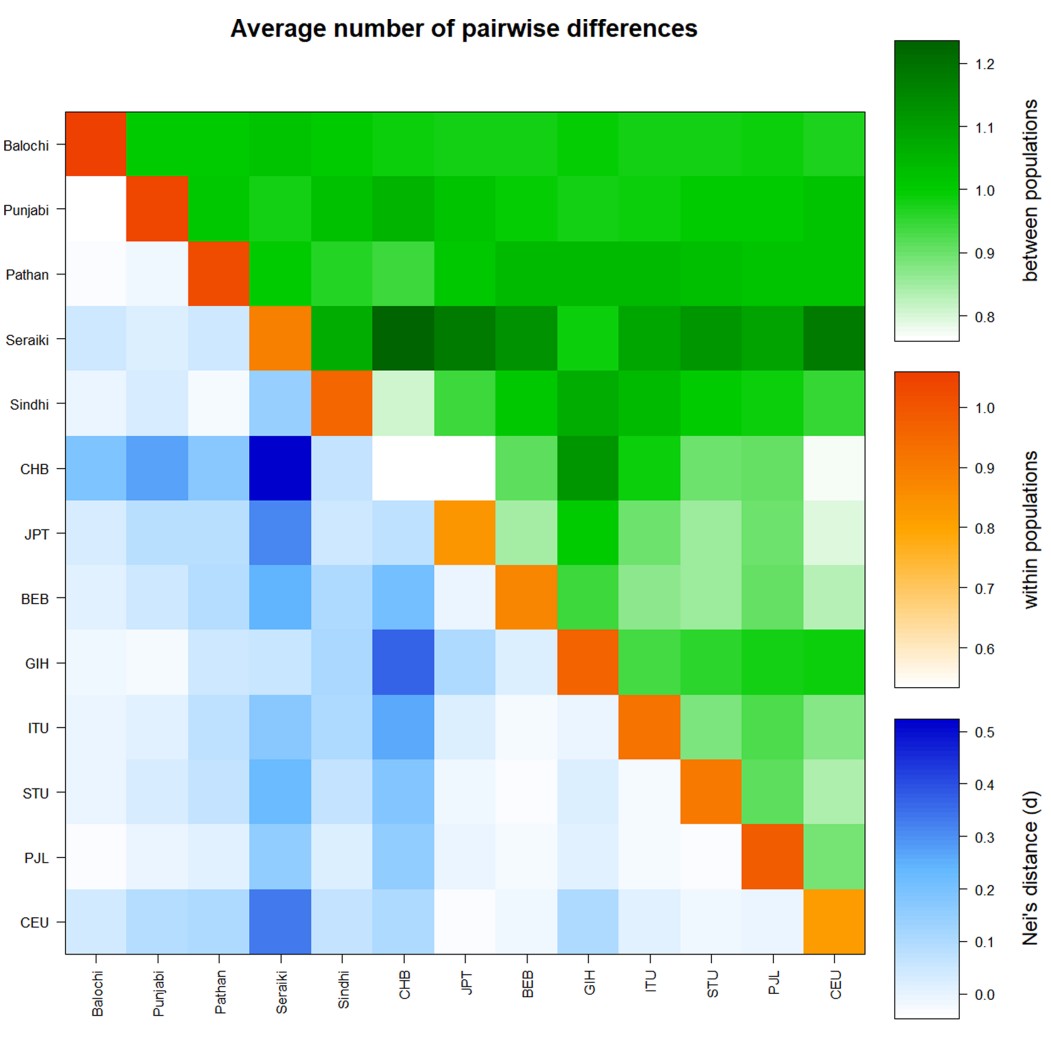

**Figure 3 Average pairwise differences-Global populations.** Pairwise comparison of Pakistani ethnic groups with other Asians populations i.e. -East Asian: BEB, GIH, ITU, STU.; South Asian: CHB, JPT and Europeans CEU. retrieved from 1000 Genomes browser, found that high genetic distinction between population (green above diagonal) of Seraiki, Sindhi as with the Nei's distance (blue below diagonal) followed by other Asian and European population.

During medium-joining haplotype networking analysis, the main haplotypes of Punjabi, Pathan, Sindhi and Seraiki ethnic groups persisted on common nodes, whereas the Balochi samples predominant haplotype-5 constituted a separate node, showing the least genetic connectivity with the rest of the ethnic groups (Fig. 4).

## DISCUSSION

The DNA sequencing approach can be implemented effectively to assess the allelic distribution of the human DNA sequences and to estimate interethnic genetic variation among populations. This approach has been frequently implemented across the world to identify specific individual polymorphisms. The Pakistani population is a heterogeneous admixture and exhibits high ethnic diversity due to complex demography and various

## Haplotypes Network

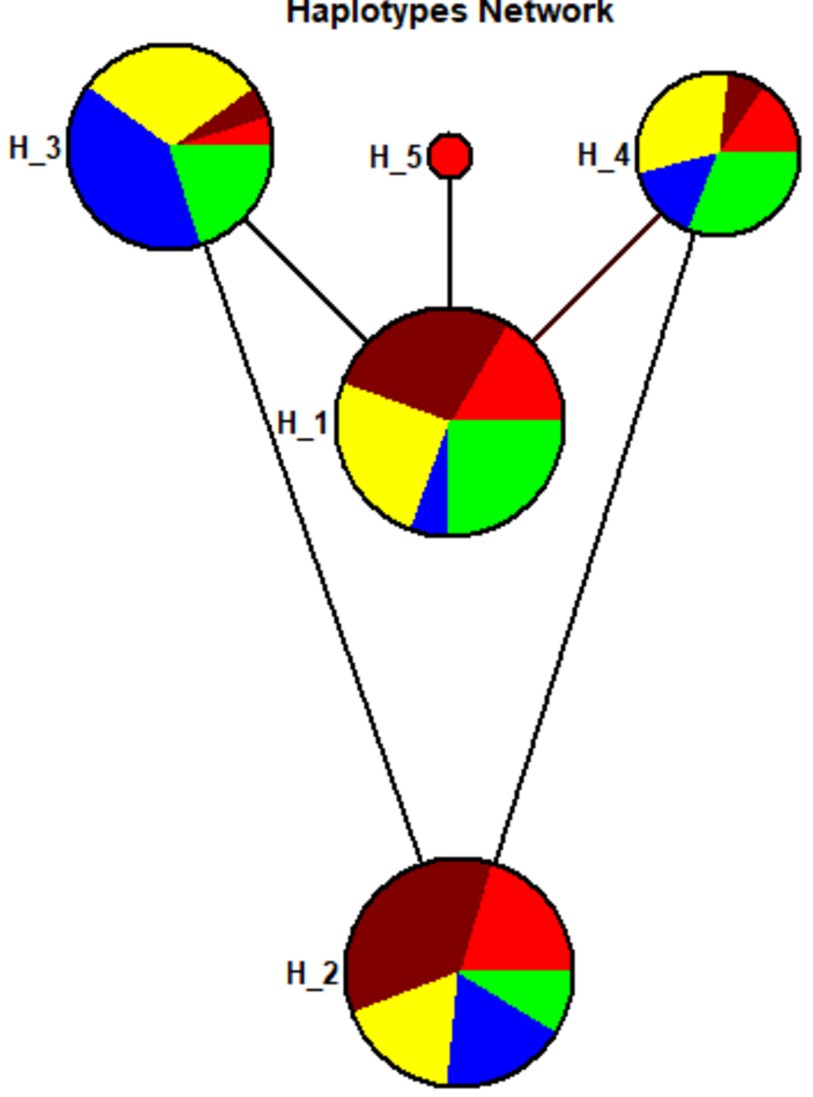

**Figure 4 Haplotypes network.** Haplotypes Network plot: Circle area are proportional to the number of taxa holding that particular haplotypes. The lines connecting the haplotypes reflect the distance of relatedness. While the colored show distinction between ethnic groups—Red: Balochi, pansy-purple: Punjabi, yellow: Pathan, blue: Seraiki, green: Sindhi.     

invasions, including Central Asian, Arabs and British colonial invasions (*Bhatti et al., 2017*). Therefore, allelic distribution-based assessment of the drug-metabolizing enzymes coding genes, including *CYP2B6* in different ethnicities of the Pakistani population is expected to be a complex but an interesting project to study.

Current study provides the genetic structure of the *CYP2B6* gene in major ethnic groups across the Pakistani population. The enzyme encoding from *CYP2B6* is responsible for the metabolism of various pharmacological drugs including efavirenz (*Ward et al., 1991*). The polymorphisms present in the *CYP2B6* gene showed a distinction of allelic frequencies and genetic variability among Pakistani ethnic groups. These differences may lead to variations in the drug-metabolizing potential of the *CYP2B6* enzyme. The former

global population genomics projects like Human Genome Diversity Project—HGDP (*Auton et al., 2015*; *Bergström et al., 2020*) identified genetic distinction among various Pakistani ethnic groups. The various Pakistani population groups (Indus-valley populations) revealing genetic patterns akin to the European, Caucasian, and Indian populations (*Metspalu et al., 2011*).

The determination of the various single nucleotide polymorphisms and allelic distribution of human genes at a population level and their association with disease phenotypes have been reported previously (*Hamblin, Thompson & Di Rienzo, 2002*; *Nyakutira et al., 2008*). Several genetic variants of *CYP2B6* are reported to have a significant association with different drug pharmacokinetics in multiple populations across the world, including African, Asian, and South American (*Klein et al., 2005*; *Nyakutira et al., 2008*; *Radloff et al., 2013*). In the current study, we observed significant genetic variation across *CYP2B6* locus in Pakistani ethnicities in comparison to worldwide, including South Asian, East Asian, and European populations. The pairwise population genetics analyses of the data inferred less genetic differentiation among Punjabi, Pathan, Balochi samples, while the Seraiki, Sindhi, and Balochi samples displayed high genetic differentiation. The highest value of pairwise genetic differences was observed between the Seraiki and Sindhi groups (Fig. 1).

Interestingly the Seraiki group showed uniform genetic composition and the lowest intra-population genetic variance (Fig. S2). Therefore, we can expect a uniform phenotype in the Seraiki population within the context of drugs metabolized by the CYP2B6 enzyme. The Seraiki group seems to be distinct from the Punjabi ethnicity, despite these groups have close demography as known from Bronze age Harappan civilization (the Pre-Aryan people) (*Ahmed & Khan, 2017*; *Shackle, 1977*; *Hashmi & Majeed, 2014*).

The Balochi ethnicity showed higher intra and interethnic genetic differentiation, although this group holds a small population size compared to the rest of Pakistani ethnicities. This heterogeneous genetic ancestry, as observed in the Balochi samples, may be due to their Aryan, Arab, Persian, Turkish, Khurdish, Dravidian, Sewais (Hindu) and black African mixed ancestries (*Ahmed & Khan, 2017*). Therefore, due to high intraethnic genetic differentiation, the Balochi ethnic group might exhibit distinct *CYP2B6* mediated drug-metabolized and treatment response phenotypes compare to other Pakistani ethnicities.

AMOVA, which was performed based on haplotypes significance analysis to determine the degree of regional differentiation and homogeneity within and among Pakistani ethnic groups, found that the majority of the genetic variance is attributed due to within-population factors. Genetic variance among populations was only 1.08%, as inferred during AMOVA analysis (Table 2). This variability was due to the intra-ethnic group heterogeneity. The genetic variance between ethnic groups in our study was low but is still considered significant as per previous studies (*Adeyemo et al., 2005*). Nonetheless, the small genetic difference between ethnic groups variance does not imply that they cannot be distinguished from each other. Instead, the pairwise analysis reveals significance differences between the ethnic groups (Fig. 1). Likewise, the results based on the haplotypes composition (Fig. S4) and haplotypes network plot (Fig. 4) among ethnic

groups is congruent to the pairwise Fst analysis and further confirmed the genetic distinction among ethnic groups.

The prevalence of *CYP2B6\*6* allele has been found variable throughout the world. Globally, G allele frequency is 0.73 and that of T allele is 0.26 (*Auton et al., 2015*). In the present study, the prevalence of T allele was found at 0.338 in the Pakistani population which is slightly lower than its frequency found in the South Asian populations. Its prevalence in East Asia and Europe was found lower than Pakistan at 0.215 and 0.236, respectively while its frequency was higher in American and African populations, i.e. 0.373 and 0.374 respectively (*Auton et al., 2015*).

If clinicians have information about the patients' *CYP2B6* gene, they may help enhance the efficacy and reduce the adverse effect by prescribing the most suitable and safest drug to the patients based on their genetic structure. Over 2.6 billion unit doses of drugs are dispensed each year in Pakistan (*Ahmed et al., 2020*) and approximately 7.2% marketed drugs are metabolized by CYP2B6 enzyme (*Zanger & Schwab, 2013*). This means that over 187 million doses of those drugs are dispensed annually in Pakistan which are metabolized by CYP2B6 enzyme. Our study shows that about one third of Pakistan's population has a *CYP2B6\*6* allele. This implies that over 62 million doses of drugs dispensed annually in Pakistan may not have desired effects as patients receiving these medications possess a low activity *CYP2B6* allele. For example, patients taking efavirenz may suffer from enhanced frequency and severity of adverse effects if they possess one or two *CYP2B6\*6* alleles. However, longitudinal studies with proper follow up are needed in the Pakistani population to confirm these findings.

## CONCLUSION

Overall, our results demonstrate distinct genetic patterns in different ethnic groups across Pakistan. The novel heterogeneous SNP, as identified in the present study, might lead to altered drug-metabolizing potential of *CYP2B6* and, therefore, additional studies are required to decipher the pharmacological impact of this SNP. The rs4803419, which has been previously shown to be associated with increased plasma efavirenz level in different populations e.g., USA, Caucasian, Black, and Hispanic (*Holzinger et al., 2012*) and Black South African populations (*Sinxadi et al., 2015*), was found in high frequency in Pakistani population. Likewise, the rs3745274 has also been previously shown in strong association with increased plasma Efavirenz level in African and Serbian populations (*Olagunju et al., 2014*; *Sinxadi et al., 2015*). This variant represents the *CYP2B6\*6* allele, which decreases its enzymatic activity (*Klein et al., 2005*). This SNP, like rs4803419, is observed in the current study for Seraiki, Sindhi, and Balochi samples with higher allelic frequencies compared to other ethnic groups. These findings provide a foundation for future studies on the mechanism and effects of *CYP2B6* polymorphisms in the Pakistani population and are a step towards personalized medicine.

## ACKNOWLEDGEMENTS

The authors wish to thank Shifa Tameer-e-Millat University, Islamabad, for providing an excellent academic environment to facilitate this kind of scholarly activity.

### Funding

This work was supported by a research grant to Dr. Sagheer Ahmed by the Shifa Tameer-e-Millat University, Islamabad. The funders had no role in study design, data collection and analysis, decision to publish, or preparation of the manuscript.

### Grant Disclosures

The following grant information was disclosed by the authors:
Shifa Tameer-e-Millat University, Islamabad.

### Competing Interests

The authors declare that they have no competing interests.

### Author Contributions

- Sagheer Ahmed conceived and designed the experiments, authored or reviewed drafts of the paper, and approved the final draft.
- Hizbullah Khan performed the experiments, analyzed the data, prepared figures and/or tables, and approved the final draft.
- Asifullah Khan analyzed the data, authored or reviewed drafts of the paper, and approved the final draft.
- Muhammad Hanif Bangash performed the experiments, authored or reviewed drafts of the paper, and approved the final draft.
- Abrar Hussain conceived and designed the experiments, authored or reviewed drafts of the paper, and approved the final draft.
- Mughal Qayum performed the experiments, analyzed the data, authored or reviewed drafts of the paper, and approved the final draft.
- Mohammad Hamid Hamdard conceived and designed the experiments, authored or reviewed drafts of the paper, and approved the final draft.

### Ethics

The following information was supplied relating to ethical approvals (i.e., approving body and any reference numbers):

Institutional Review Board & Ethics Committee of Shifa Tameer-e-Millat University and Shifa International Hospital, Islamabad approved this research (IRB#990-265-2018).

### Data Availability

DNA sequencing data generated in this study are available at GenBank: MW017225–MW017328.

### Supplemental Information

Supplemental information for this article can be found online at http://dx.doi.org/10.7717/peerj.11149#supplemental-information.

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
