# Peer review of "Inter-ethnic genetic variations and novel variant identification in the partial sequences of CYP2B6 gene in Pakistani population"

_PeerJ, doi:10.7717/peerj.11149_

## Round 0.1 · original submission · Major Revisions

Your manuscript was considered interesting and valuable, but one of the reviewers raised important comments that need to be addressed. Specifically, the reviewer had concerns about stating that the entire CYP2B6 gene was studied, when only two Single Nucleotide Polymorphisms were analyzed. Additionally, the reviewer stated that putting your results for allele *6 in the context of the frequency of this allele for other ethnic groups is important.

Please, submit a detailed rebuttal which shows where and how you have taken all comments and suggestions into consideration. If you do not agree with some of the reviewers’ comments or suggestions, please explain why. Your rebuttal will be critical in making a final decision on your manuscript. Please, note also that your revised version may enter a new round of review by the same or by different reviewers. Therefore, I cannot guarantee that your manuscript will eventually be accepted

Reviewer 1 ·

Basic reporting

The work seems very important to me because it is an original article on different ethnic groups, which shows important contributions due to inter-ethnic differences in the same Pakistani population, showing the decrease in drug activity of two ethnic groups of the same population (Seraiki and Barochi); However, it is important that in the paragraphs where the activation and inactivation of drugs are mentioned, it is very important that they indicate which drugs (at least 3 to 5 drugs)

Experimental design

It is very important to clarify whether the informed consents obtained in the different ethnic groups were presented and obtained in the language of each ethnic group, given that in the same country not all their ethnic groups speak the same language.

Validity of the findings

Although the polymorphic differences of CYP450 cytochrome among the 5 Pakistani ethnic groups are evident, it is very important that they point out the differences in terms of the regions of the country where they live, customs, type of diet, habits, etc.

Additional comments

It is very important that you complete the responses to the previously suggested comments.
Respect work represents a valuable contribution

·

Basic reporting

I consider that the title of this work is very broad for the true scope of this research, since a study is not carried out on the complete CYP2B6 gene, only a region of it.

The background of the abstract is very ambiguous. I suggest targeting your reach directly to rs4803419 and rs3745274. Since in methods it is directly pointed out that a region containing these important SNPs is sequenced.

Introduction
- When referring to drugs metabolized by CYP2B6, it is necessary to include antiretrovirals (efavirenz).
- Paragraph of lines 67 to 69: there is a serious error in the wording, since it assumes that any SNPs in the CYP2B6 gene would affect the expression and/or activity of the enzyme.
- line 69: it should not refer to "races", it can speak of "ethnicities".
- line 71 and 76: improve the writing, consider that an allele is a variant of the gene.
- lines 81 to 82: it is not understood what it refers to.
- line 85: Please verify the indicated information, rs2279343 is not c.516G>T, but c.785A>G. For c.516G>T corresponds rs3745274.

It is necessary to add in the introduction the main characteristics of the ethnic groups of the Pakistani population. Since this is the main rationale for the research question.


Material and methods
- Informed consent is not just a written and signed document, it is a process.
- lines 112 to 113 should be moved to the introduction.
- line 131, you should improve the writing (twice "analysis")


Results

- In the PDF appear many: "Error! Reference source not found".
- Allele frequencies are better expressed as decimals not as a percentage (100% = 1.0)
- Table 2 is not referenced in the results.
- Verify that all supplementary figures or tables are referenced in the document.

Table 1

- For the position of the SNP, you must indicate which is the name of the sequence (for example: GRCh38.p12 chr 19)
- The information in the "Position" column is repetitive with the "Effects" column, since apparently only the reference sequence changes ...?
- On the other hand, information is repeated from the "REF" column and the "Effect" column (both indicate the nucleotide change ...?).
- At the bottom of the table include the meaning of REF and ALT.

The foot of table 1, seems to correspond to table 2.?

The wording needs to be improved.

Experimental design

Material and methods

The pattern observed in a 1% agarose gel allows to evaluate its quality (integrity), but does not allow to quantify the DNA (only a semi-quantification). Quantification should be by spectrophotometric or fluorescence methods.

Validity of the findings

A comparative analysis of the frequencies observed for this allele (*6) with other ethnic groups worldwide is necessary.

The discussion and conclusions are quite adequate, however, they require better support from the introduction and the title of the paper.

On the other hand, I consider that the observation of a "new" SNP should be indicated in the title.

Additional comments

It would be very interesting to predict what percentage of the Pakistani population (or ethnic groups studied) would be poor metabolizers, based on the observed genotypes, and discuss their differences or similarities with other ethnic groups or populations globally.

I recommend seeking information on adherence to antiretroviral treatment with efevirez in the Pakistani population. It would be interesting to see how the rates of adverse events to this drug and/or discontinuation of therapy are. Then correlate with the observed frequencies for the *6 allele, compared to other ethnicities or populations.

---

## Round 0.2 · accepted · Accept

You have addressed the reviewers' comments in a thorough and satisfactory fashion.

Reviewer 1 ·

Basic reporting

Clear and unambiguous, professional English used throughout


Literature references, sufficient field background/context provided.


Professional article structure, figures, tables. Raw data shared

Self-contained with relevant results to hypotheses

Experimental design

Original primary research within Aims and Scope of the journal.

Research question well defined, relevant & meaningful. It is stated how research fills an identified knowledge gap.


Rigorous investigation performed to a high technical & ethical standard.


Methods described with sufficient detail & information to replicat

Validity of the findings

All underlying data have been provided; they are robust, statistically sound, & controlled.


Conclusions are well stated, linked to original research question & limited to supporting results

Additional comments

I found it to be an excellent job, very well prepared and developed